# Learning Long-Term Crop Management Strategies with CyclesGym

**Matteo Turchetta**
ETH Zurich
Zurich, Switzerland
`matteotu@ethz.ch`

**Luca Corinzia**
ETH Zurich
Zurich, Switzerland
`lucac@ethz.ch`

**Scott Sussex**
ETH Zurich
Zurich, Switzerland
`ssussex@ethz.ch`

**Amanda Burton**
Agroscope
Nyon, Switzerland
`amanda.burton@agroscope.admin.ch`

**Juan Herrera**
Agroscope
Nyon, Switzerland
`juan.herrera@agroscope.admin.ch`

**Ioannis N. Athanasiadis**
Wageningen University & Research
Wageningen, Netherlands
`ioannis.athanasiadis@wur.nl`

**Joachim M. Buhmann**
ETH Zurich
Zurich, Switzerland
`jbuhmann@ethz.ch`

**Andreas Krause**
ETH Zurich
Zurich, Switzerland
`krausea@ethz.ch`

## Abstract

To improve the sustainability and resilience of modern food systems, designing improved crop management strategies is crucial. The increasing abundance of data on agricultural systems suggests that future strategies could benefit from adapting to environmental conditions, but how to design these adaptive policies poses a new frontier. A natural technique for learning policies in these kinds of sequential decision-making problems is reinforcement learning (RL). To obtain the large number of samples required to learn effective RL policies, existing work has used mechanistic crop growth models (CGMs) as simulators. These solutions focus on single-year, single-crop simulations for learning strategies for a single agricultural management practice. However, to learn sustainable long-term policies we must be able to train in multi-year environments, with multiple crops, and consider a wider array of management techniques. We introduce CYCLESGYM, an RL environment based on the multi-year, multi-crop CGM Cycles. CYCLESGYM allows for long-term planning in agroecosystems, provides modular state space and reward constructors and weather generators, and allows for complex actions. For RL researchers, this is a novel benchmark to investigate issues arising in real-world applications. For agronomists, we demonstrate the potential of RL as a powerful optimization tool for agricultural systems management in multi-year case studies on nitrogen (N) fertilization and crop planning scenarios.

## 1 Introduction

The global food system is under increasing pressure from several fronts. First, it must meet the caloric demand of the world population, which is projected to reach 9.8 billion by 2050 [59], despite constrained resources like fresh water and arable land. Second, our agricultural system must adapt to more frequent and extreme weather events induced by climate change [24]. At the same time, agriculture has a profound environmental impact. For example, Nitrogen (N) is fundamental for plant growth, but its production in the form of mineral fertilizer and its incomplete uptake by plants

36th Conference on Neural Information Processing Systems (NeurIPS 2022) Track on Datasets and Benchmarks.

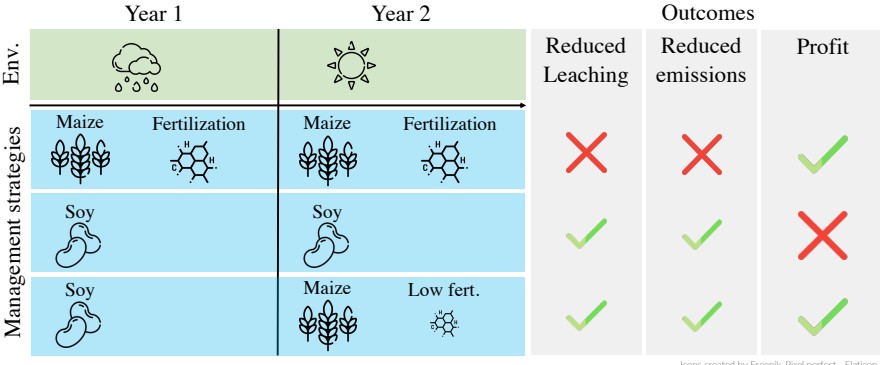

Figure 1: Agricultural management is a long-term problem with interdependent decisions. In this example, the top management strategy attains high profits by using maize with non-organic fertilization but it has a high environmental impact due to leaching and greenhouse gas emissions caused by the fertilizer. The middle strategy has a low environmental impact but lower profits due to the reduced yields of soybeans. The bottom strategy attains high profits but has a lower environmental impact compared to monoculture maize by leveraging the nitrogen fixation of soybeans to obtain greater maize yields with smaller amounts of fertilizer.

are major causes of greenhouse gas emissions [38], surface and groundwater pollution [36]. Other agricultural practices can also have negative impacts, e.g., reduction in soil organic matter from tillage [27], aquifer depletion from irrigation [52], and more.

To address these challenges, it is necessary to design efficient and sustainable crop management strategies that adapt to environmental conditions. This is becoming more practical due to the increasing availability of data on agricultural systems [3, 21]. In domains such as games and robotics, a very successful tool to learn adaptive strategies from environment feedback has been (deep) reinforcement learning (RL) [57]. Unfortunately, RL's notorious sample inefficiency makes it impossible to apply directly to agriculture, where turn-over times are prohibitively long. To overcome this, [43, 22] propose RL environments for crop management built on mechanistic crop growth models (CGMs). These environments simulate a single year [43, 22], a single type of crop management [43], and a single crop [22]. However, crop management is a problem with a long time horizon and multiple interdependent decisions, see Fig. 1. To develop RL algorithms applicable in the real world, we need benchmarks that capture the specific complexities of agricultural problems.

We introduce CYCLESGYM, the first crop-management RL environment that facilitates the learning of multi-year strategies with complex action spaces and multiple crops. While we provide a set of ready-to-use scenarios, the modular nature of CYCLESGYM allows users to compose action and observation spaces, combine rewards and costs, and define weather and soil generating modules to create custom environments. For RL researchers, this is a novel benchmark that offers the possibility to investigate relevant issues arising in real-world RL while tackling pressing societal problems. For agronomists and environmental scientists, CYCLESGYM allows users to leverage advances in RL to improve the management of agricultural systems. We demonstrate this potential in long-term N fertilization and crop planning simulations that are not possible in existing RL environments. There, our RL agents outperform expert practices during training and in a variety of unseen test scenarios.

Environments like CYCLESGYM are important for data-driven agriculture as they allow to investigate ideas that would be prohibitive in reality. In the future, their relevance is likely to increase as improvements in CGMs [33] and their calibration [1] are narrowing the gap between simulation and reality (sim2real). However, this gap is larger in agronomic systems than in typical RL applications and extra care is necessary when extrapolating results obtained in CYCLESGYM to the real world.

## 2 Related Work

**Machine learning in agriculture**. Data-driven applications in agriculture evolve quickly, as more experience, applications, and computational power become available [42]. Machine learning (ML) proves useful to address a variety of tasks from yield forecasting [45], digital phenotyping

and breeding [61], weed detection and crop protection [62], and a large variety of crop, water, soil, and livestock management tasks [10]. To utilize ML, multiple data sources are employed, including remote-sensed imagery from satellites, and unmanned aerial vehicles, typically fused with meteorological data, along with additional environmental information and management actions collected on farms. While the number of agricultural applications of machine learning is growing rapidly [10], the domain suffers from a lack of standardized data and benchmarks.

**Crop growth models**. CGMs are mechanistic models that simulate crop growth driven by the environment and management practices [64]. Different CGMs may differ significantly in their strengths, weaknesses and functionality (a thorough comparison is beyond the scope of this work, see [30, 19] for extensive reviews). Therefore, which CGM to use depends on the research question under investigation and no single best one is available.

Cycles [33] (used in this study) is adept at capturing the long-term effects of management practices and at representing complex agricultural systems and the resulting interactions. This includes modeling multi-year rotations [46], which remains difficult in other CGMs (for more details, see Section 5). Additionally, Cycles has been shown to simulate soil carbon-nitrogen dynamics accurately [46], which makes it suitable for studying N management. Another widespread CGM is APSIM, which has been continuously in development since the early 1990s [37]. As such, users introduced modules to simulate many crops, including less common ones. DSSAT [31] is also widely used and can additionally be coupled with groundwater models like HYDRUS [54], to consider further environmental outcomes. WOFOST [60] simulate plant growth response to increasing atmospheric $CO_2$ concentrations and hence has been used extensively for climate change adaptation studies [41].

**Crop management with RL**. The potential of RL for crop management optimization has been advocated strongly in the last decades. For example, [7, 13] propose a framework to use RL in this context. However, these are conceptual works and do not offer concrete implementations.

In highly controlled environments, promising results have been attained. For example, the autonomous greenhouse challenge [28] proposes a competition for automatic control of a smart greenhouse where RL-based strategies outperform human experts. The work in [5] uses a neural network to predict the crop state in a greenhouse and uses RL to optimize its management. The authors of [15] presents a similar approach that validates results in real-world trials.

Attaining similar results in open-field agriculture is more challenging as both sensing and actuation capabilities are reduced. In this context, [11] compares dynamic programming and Q learning [65] to optimize the profitability of an irrigation policy based on the MODERATO model [12]. Similarly, [6] optimizes the irrigation of potatoes using the SIMPLE [68] model. However, such models fail to simulate important dynamics. In contrast, [56] learns an irrigation policy from a surrogate model trained on DSSAT [31]. This study is limited to a tabular algorithm with 20 states and 4 actions. A more recent work [43] uses CropGym, an OpenAI gym [14] wrapper around WOFOST [60], to optimize N fertilization with PPO [53]. [66] solves a similar problem using gym-DSSAT [22] in two locations with SAC [25] and deep Q learning [39]. Moreover, it studies the effect of partial information and reduced temporal resolution. Concurrently to our work, [23] studied the trade-offs between yield, nitrogen use efficiency, and leaching in fertilization experiments with gym-DSSAT. Moreover, they presented irrigation experiments in the same environment, demonstrating its ability to simulate multiple management practices. While these works are closely related to ours as they use deep RL with sophisticated CGMs, they are limited to fertilization management in 1-year scenarios with a single crop.

# 3 Background

This section introduces the formalism of RL, i.e., Markov decision processes (MDPs), partially observable Markov decision processes (POMDPs), and constrained Markov decision processes (CMDPs).

**MDPs**. Markov decision processes (MDPs) [47] formalize sequential decision-making problems where an agent observes the state of its environment and takes actions to control it and maximize rewards. They are described by the tuple $\langle \mathcal{S}, \mathcal{A}, \mathcal{P}, r, \gamma \rangle$. $\mathcal{S}$ and $\mathcal{A}$ are the state and action spaces, respectively. $\mathcal{P}(s'|s,a)$ and $r(s,a,s')$ are the transition kernel and the reward function. They describe the probability of reaching state $s'$ when the agent takes action $a$ in $s$ and the reward obtained by the agent in the process. Finally, $\gamma \in [0,1]$ is the discount factor determining the relative value of present and future rewards. The goal of an RL agent is to find a policy, *i.e.*, a mapping from

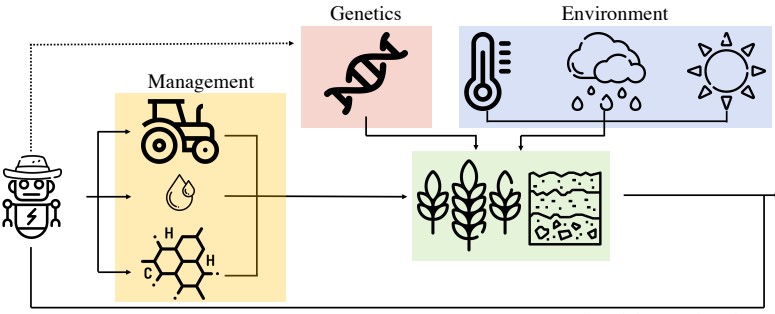

Figure 2: In the general crop optimization problem we want to make managerial (M) and genetic (G) decisions to optimize desired crop-related quantities in an uncertain environment (E). The genetic decisions are denoted by a dashed line as they are not investigated in CYCLESGYM.

states to distributions over actions, $\pi : \mathcal{S} \to \Delta_{\mathcal{A}}$, that maximizes the expected sum of discounted rewards, $\pi^* \in \arg\max_{\pi \in \Pi} \mathbb{E}_{\rho^\pi} \left[ \sum_{t=0}^{T} \gamma^t r(s_t, a_t, s_{t+1}) \right]$, where $\rho^\pi$ is the distribution of the states and actions induced by the policy $\pi$ for a fixed initial state distribution, and $T$ is the time horizon.

**Partial observability**. In MDPs, we assume the agent has perfect knowledge of the system state. POMDPs [40] are a generalization of MDPs to cases where the agent has access to incomplete information. They are described by a tuple of the form $\langle \mathcal{S}, \mathcal{A}, \mathcal{P}, r, \Omega, \mathcal{O}, \gamma \rangle$, where, in addition to the standard MDP elements, we introduce $\Omega$ as the space of possible observations and $\mathcal{O}(o|s', a)$ as the probability of obtaining observation $o$ when we reach $s'$ taking action $a$. In this case, the policy is a mapping from the observation history to a distribution over actions.

**Constraints**. MDPs and POMDPs are unconstrained optimization problems. CMDPs [4] are a generalization of MDPs augmented with $m$ cost functions $c_1, \ldots, c_m$, where $c_i(s, a, s')$ is the $i^{th}$ cost incurred when transitioning from $s$ to $s'$ taking action $a$. In CMDPs, the goal is to find the policy that maximizes the return within a set of candidate policies that keep the expected sum of discounted costs below 0, i.e., $\Pi_C = \{ \pi \in \Pi : E_{\rho^\pi} \left[ \sum_{t=0}^{T} \gamma^t c_i(s_t, a_t, s_{t+1}) \right] \leq 0, \forall i \in [1, m] \}$.

## 4 Problem statement

Crop optimization is a complex problem influenced by genetic, environmental, and managerial factors (see Fig. 2). In open-field agriculture, control over the environment is limited. Therefore, the space of possible decisions spans genetic and managerial ones. Since modeling the interplay between these two is a challenging open problem, in this work we focus on management optimization for fixed genetics.

Section 4.1 frames the management of agricultural systems in the context of MDPs and their extensions. Section 4.2 introduces some RL research questions arising when optimizing crop management strategies.

### 4.1 RL for the management of agricultural systems

To optimize agricultural management with RL, we need to express this problem in the language of (PO)MDPs, i.e., we must specify the $\langle \mathcal{S}, \mathcal{A}, \mathcal{P}, r, \gamma \rangle$ tuple. Here, we present the general formulation of this problem. In Section 5, we describe how it is implemented in CYCLESGYM.

**States and observations**. In MDPs, states are assumed to be Markovian, i.e., to contain all the information necessary to predict future states given actions. In agriculture, these variables include soil and crop states and weather variables. However, some variables, e.g., soil composition, may not be available without expensive soil surveys. To model this, we can use the partial observations of POMDPs.

**Actions**. The action space contains all the decisions involved in agricultural management, including fertilization, irrigation, planting, and tillage. Each of these macro-actions has several degrees of freedom, which renders the action space complex. For example, a fertilization decision is determined by the timing, the total amount of fertilizer, the relative concentration of nutrients, and more.

**Transitions**. In agriculture, transition probabilities are determined by the weather and the processes taking place within the crop and in the soil. In simulation-based RL for agriculture, these are approximated to different degrees of accuracy by CGMs.

**Reward**. Many options are possible to specify the reward. For example, [66] use linear combinations of different quantities, including yield, nitrate leaching, and fertilization penalization. However, these quantities are incommensurable, and setting the linear coefficients is not intuitive. Similar to [56, 11], we use profitability as a reward, as it ultimately drives the adoption of management practices.

**Constraints**. While profitability is an intuitive objective, it disregards important aspects of the problem, e.g., environmental norms. We believe that adding constraints, as modeled by CMDPs, can be a more principled way of modeling these requirements than incorporating negative rewards.

## 4.2 RL research questions

This section presents interesting challenges for RL research posed by the problem above.

**Context**. RL agents base their decisions on the information provided by observations. In agriculture, causal domain knowledge is available that allows us to treat some of this information differently and reduce the effective size of the observation space. Exploiting this may improve sample efficiency and generalization outside of the simulation. For example, the agents' actions affect the soil and the crop but they do not influence the weather, which is better captured by a context variable [9, 26]. Other relevant contexts may include the market price of materials, e.g., fertilizers and seed costs.

**Feature selection and observation cost**. In real-world agriculture, obtaining information is expensive and time-consuming. The benefit of acquiring certain data may not outweigh the cost. Therefore it is important to carefully select which variables to measure. We distinguish between two kinds of costs: one-time, where we buy a sensor and continuously get measurements, *e.g.*, temperature, and repeated, where we pay each time we get a measurement, *e.g.*, soil analysis. For one-time costs, Bayesian optimization methods [18] can be used. For repeated costs, [8] studies the case where agents can obtain extra information at a cost.

**Transfer learning**. The performance of agricultural practices depends on the environment where they are deployed. For example, the impact of fertilization decisions may vary depending on the soil condition. Unfortunately, it may not be possible to separately train a policy for every possible environment. However, transfer learning [69] and meta-learning [32, 50] methods can be used to efficiently adapt existing policies to new environments.

**Interpretability**. To promote the adoption of smart solutions in agriculture, it is important to understand why RL agents make certain decisions to confront them with best practices and domain knowledge. In this regard, it is possible to use policy distillation [51] to summarize complex neural network policies with more interpretable models or directly train interpretable policies like in [44].

**Time scales**. In standard RL, agents make decisions at a uniform time scale. In agriculture, this is not the case. For example, we can choose the crops to plant yearly. However, when to plant the crop within the year is a decision that depends on the temperature and soil condition that must be taken on a daily scale. To address this, methods from hierarchical RL [58, 63] could be investigated.

**Sparse actions**. In common RL tasks, e.g., robotics, policies are dense in the sense that actions must be taken almost every time step. In contrast, in agriculture, we know a priori that a strong agent should act sparsely, i.e., should be idle most of the time and rarely take action. Besides using constraints, we can encourage sparsity by using regularizers like in [67].

**Sim2Real gap**. CGMs can show a significant gap between the predictive and the true crop dynamics if used out-of-the-box on real-world data. To address the gap, practitioners calibrate the model parameters with existing data [1]. A complementary approach is the development of RL algorithms that are robust to distributional shifts between true and simulated dynamics. While this problem has been successfully addressed in robotics [2, 29], new algorithms may be necessary in the agronomic domain, where distributional shifts may be considerably larger.

**Offline RL**. Directly applying RL in real-world agriculture may be prohibitive due to the long turnover times and high stakes of these scenarios. However, the abundance of historical data collected with (possibly) sub-optimal policies makes offline RL [35] a promising method in this context.

# 5 CYCLESGYM

CYCLESGYM is an OpenAI-Gym-compatible environment [14] that relies on the Cycles CGM. It provides predefined environments and allows users to flexibly and easily create custom ones. The code, user manual, and tutorials are available as a *public* repository at `https://github.com/kora-labs/cyclesgym`.

**Cycles**. Cycles is a mechanistic multi-year and multi-species agroecosystem model [33], which evolved from C-Farm [34] and shares biophysical modules with CropSyst [55]. It simulates the water and energy balance, the coupled cycling of carbon and nitrogen, and plant growth at daily time steps. Its ability to simulate a wide range of perturbations of biogeochemical processes caused by management practices for multiple crops and its focus on long-term simulations make it a suitable CGM to study the application of RL to sustainable agriculture, where these aspects are crucial (see Fig. 1).

**CYCLESGYM structure**. Cycles's simulations are controlled via configuration files and their results are written to output text files. CYCLESGYM interacts with Cycles via managers that allow it to parse, edit, and save such files in Python. These managers are grouped in classes, known as observers, implementers, rewarders, and constrainers, that provide an interface to compute observations, implement actions, give rewards, and enforce constraints respectively. By combining such classes, it is possible to create custom observation spaces and rewards. The generic environment `CyclesEnv` performs the operations that are common to all environments, *e.g.*, initializing dedicated files, launching simulations, and parsing outputs. By sub-classing `CyclesEnv` and overriding its methods that instantiate the observers, rewarders, constrainers, and implementers, it is possible to create a wide range of custom RL environments. Moreover, CYCLESGYM lets users specify custom scenarios by providing soil and weather files or it allows generating these files at run time with custom weather and soil generative models. Finally, it allows for simulation with a large number of pre-specified plant species or custom ones described by user-specified crop parameters.

**Pre-defined environments**. CYCLESGYM's focus is on its flexibility to create custom RL environments for agricultural management in a modular fashion. Nonetheless, we provide a series of pre-defined environments for N fertilization and crop planning varying along multiple axes:

- *Location*: Cycles natively ships with weather data corresponding to two locations in Pennsylvania, Rock Springs and New Holland. We provide all our environments for both locations.
- *Weather*: In our environments, we allow the weather to be fixed for the year that is being simulated or to be randomly sampled from the available historical data at the beginning of each simulation.
- *Duration*: For the fertilization environments we provide a short, a medium, and a long-term version of the environment lasting 1, 2, and 5 years, respectively. For the crop planning environment, we only provide one scenario as this problem is meaningful only over long horizons.
- *Constraints*: For the fertilization environments, we provide a constrained and an unconstrained version. In the constrained version, it is possible to limit the number of fertilization events, the total amount of fertilizer applied, N leaching, N volatilization, and N emission. All constraints are implemented with the SAFETYGYM [49] interface which is compatible with algorithms designed for the standard OpenAI Gym interface [14]

# 6 Experiments

We present long-term experiments on two types of crop management practice: N application and crop rotation. In contrast, previous works applying deep RL to sophisticated CGMs for open field agriculture [43, 66] are limited to N application in the single-year setting. In all our experiments, we learn RL policies for a specific aspect of crop management, while keeping other practices fixed.

We train all RL agents with PPO [53] as implemented in the Stable-Baselines3 library [48]. We think that using out-of-the-box RL algorithms is insufficient to resolve many of the challenges discussed in Section 4. However, it is sufficient for demonstrating the functions of CYCLESGYM whilst answering some new questions about the capabilities of RL in this setting.

In our experiments, we compare policies trained with RL to baselines representing current practices. Clearly, these comparisons are limited to the performance in Cycles simulations. We do not make conclusions about the possible real-world performance of our learned policies due to the sim2real gap. All experiments involve growing maize or soybeans at two sites in Pennsylvania. We provide further

Table 1: Results for the RL experiments in the N fertilizer *train* scenario. We provide rewards in k$/(year·ha) for agents and baselines on the *train* setup over different time horizons. RL-1 describes an RL policy that is only trained on a one-year time horizon. The results are reported as mean (standard deviation of mean estimate). For RL, RL-1 and NONADAPTIVE, the standard deviation is estimated by performing 20 runs with different seeds and computing a bootstrap estimate. For baseline policies, we perform 100 repeats and estimate the standard deviation by applying the central limit theorem.

| Method | Horizon [years] | | |
| | 1 | 2 | 5 |
|---|---|---|---|
| **RL** | 2.05 (0.01) | 2.03 (0.01) | 2.03 (0.00) |
| NONADAPTIVE | 2.04 (0.02) | 2.01 (0.01) | 1.99 (0.01) |
| CYCLES | 1.87 (0.02) | 1.66 (0.01) | 1.56 (0.00) |
| PA | 1.86 (0.02) | 1.65 (0.01) | 1.57 (0.00) |
| ZERON | 0.74 (0.01) | 0.47 (0.00) | 0.38 (0.00) |
| **RL-1** | 2.05 (0.01) | 1.93 (0.02) | 1.74 (0.06) |

information about our experiments, including details on the crops, the locations, the hyperparameters, and everything necessary to reproduce them in the appendix. Scripts to reproduce the experiments are available at https://github.com/kora-labs/cyclesgym.

## 6.1 Fertilization

In our fertilization experiments, the RL agent observes a subset of variables related to the crop, soil, and weather, then sets the amount of N to apply to the crop on a weekly basis. We discretize the action space into 11 uniformly spaced amounts from 0 to 150 kg/ha. The agent's goal is to maximize profit per hectare, computed as the value of the crop at harvest minus the cost of N used.

Our experiments investigate the training performance and the generalisation of learnt policies. We train agents in 1, 2, and 5 year environments on weather data uniformly sampled from 1980 to 2005 for Rock Springs, Pennsylvania (this setting is denoted *train*). We denote the resulting policy with RL. We test generalization of agents across time (*test-time*), location (*test-location*), and planning horizon (*test-horizon*). For *test-time*, we test agents on weather data uniformly sampled from 2006 to 2015 in Rock Springs. For *test-location*, we test on weather data uniformly sampled from 1980 to 2005 for New Holland, Pennsylvania. For *test-horizon*, we test agents trained in a single-year setting on a multi-year environment that is otherwise identical to the *train* setting.

We consider four non-adaptive baselines. CYCLES is a strategy included in the Cycles software. PA is a policy based upon fertilization advice in [20], a source focused on Pennsylvania. Both CYCLES and PA apply N in fixed amounts during fixed weeks of each year. The ZERON strategy never applies N. It serves as a baseline to measure the contribution of weather and soil conditions to the yield. NONADAPTIVE is a policy trained with RL that observes only the date and the amount of N added to date and has no system feedback. NONADAPTIVE agents are trained and evaluated as RL agents.

Table 1 shows RL greatly outperforms the expert recommendations of PA and CYCLES in *train* achieving gains in average yearly profit ranging from 11% over one year horizon experiments to almost 30% over 5 years. These large gains are likely due to the fact that the expert policies consider additional criteria, such as environmental norms, that RL agents are not aware of. Nonetheless, they highlight the potential of smart fertilization agents. We also find a modest improvement in RL over NONADAPTIVE. This suggests that access to more information can improve profitability, albeit mildly. This is in contrast with the findings in [66], where policies with access to partial information perform considerably worse. Finally, we find that the performance of an RL agent trained for a 1 year horizon deteriorates considerably when deployed in 2 and 5 year environments, which highlights the importance of multi-year training environments supported by CYCLESGYM.

Tables 2 and 3 (in the appendix) show similar results on *test-time*, while, in *test-location*, RL has more difficulty generalizing and mildly underperforms some of the baselines. This is reasonable as the distributional shift caused by different locations may be larger than the one caused by different years.

Fig. 3 visualizes a deterministic 5-year RL policy, which selects the action assigned highest probability by the agent, on a 5 year weather sequence from *test-time*. The policy fertilizes almost entirely during the growing season, and exhibits behaviour varying across different years. Like

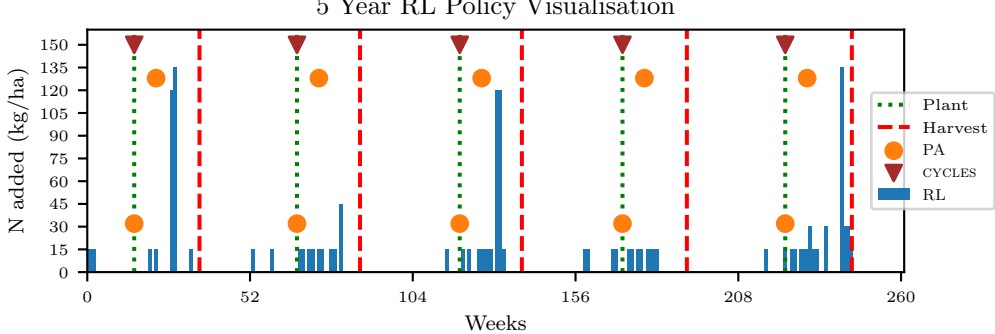

Figure 3: The amount of N added each week for a 5-year policy trained using RL and evaluated on the *test-time* setting. Vertical lines denote plant and harvest dates. The actions of static baselines (denoted CYCLES and PA) are denoted by points.

Table 2: We provide rewards in k\$/ha for agents and baselines on the *test-time* setup over different fixed costs of nitrogen fertilization, given also as k\$/ha. The results are reported similarly to Table 1.

| Fixed costs [k\$/ha] | 0.01 | 0.02 | 0.03 | 0.04 | 0.05 |
|---|---|---|---|---|---|
| **RL** | 2.00 (0.02) | 1.93 (0.02) | 1.92 (0.02) | 1.82 (0.07) | 1.87 (0.02) |
| **NONADAPTIVE** | 2.01 (0.03) | 2.02 (0.01) | 1.92 (0.11) | 2.00 (0.02) | 1.97 (0.01) |
| **CYCLES** | 1.86 (0.02) | 1.85 (0.02) | 1.84 (0.02) | 1.83 (0.02) | 1.82 (0.02) |
| **PA** | 1.84 (0.02) | 1.82 (0.02) | 1.80 (0.02) | 1.78 (0.02) | 1.76 (0.02) |
| **ZERON** | 0.74 (0.01) | 0.74 (0.01) | 0.74 (0.01) | 0.74 (0.01) | 0.74 (0.01) |

for many RL and NONADAPTIVE policies, this policy adds substantially more N than the baselines. Unlike the fixed baselines, the RL policy fertilizes often but sometimes in small amounts. This is likely because small but regular N applications may result in a greater proportion of N being taken up by the plant. Visualizations of other trained policies are in the appendix.

In the appendix, we present experiments training on fixed weather patterns, as is done in [66].

### 6.1.1 Fixed costs for fertilization applications

The policy in Fig. 3 fertilizes more often than most farmers would in practice. This is because the reward does not account for fixed costs (e.g., machines and labour) of fertilization operations. Since these costs depend on several factors including fuel costs, farm size, and opportunity costs, they are not easy to quantify. Therefore, here we study how our RL policies vary as a function of fixed fertilization costs, which we add to the variable cost of N presented above. We train RL and NONADAPTIVE agents using this new reward function in a single-year environment.

Table 2 reports the return obtained with this reward as a function of fixed costs varying between 10 and 50 \$/ha. Predictably, the new cost diminishes the return of all policies. However, both RL and NONADAPTIVE consistently outperform all baselines for all fixed costs considered in this single-year scenario. We expect larger gains for longer horizons in line with the results in Table 1. Figure 4 shows two examples of learnt policies by NONADAPTIVE (left) and RL (right), for a fixed cost of 50\$/ha. In this experiment, the learnt policies fertilize on fewer occasions (and are thus more realistic) compared to Figure 3, where no fixed fertilization cost is considered. The policy learnt by the NONADAPTIVE agent typically fertilizes fewer times than the RL one. This is also reflected quantitatively in Table 2, where NONADAPTIVE performs better than RL for increasing fixed costs. We speculate that the NONADAPTIVE agent is less sensitive to outliers in the observations (e.g. weather) compared to the RL approach, which allows it to learn more robust fertilization strategies with fewer events. Further analysis of this finding is left for future work.

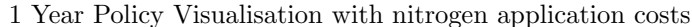

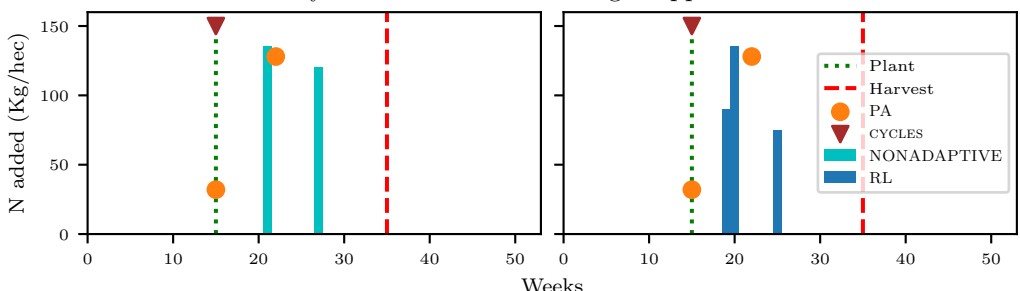

Figure 4: The amount of N added each week for a 1-year fertilization policy trained using RL and evaluated on the *test-time* setting with fixed fertilization cost equal to 50 $/ha.

Table 3: Results of the RL experiments in a crop planning scenario. We provide the average yearly profit of baselines and RL agents in different environments in k$/(year·ha). The results are reported as mean (min, max) over 5 training runs with different random seeds.

| Method | train | test-time | test-location | test-time-loc-horizon |
|---|---|---|---|---|
| **SSM** | 1.28 | 1.27 | 1.23 | 1.19 |
| **S** | 1.30 | 1.26 | 1.16 | 1.16 |
| **RL** | 1.35 (1.35,1.36) | 1.29 (1.24,1.32) | 1.18 (1.04,1.22) | 1.13 (0.74,1.25) |
| NONADAPTIVE | 1.36 (1.36,1.36) | 1.31 (1.30,1.32) | 1.24 (1.21,1.25) | 1.12 (1.03,1.20) |

## 6.2 Crop planning

Crop sequences are a common farm management practice that leverages nitrogen-fixing crops, e.g., soybeans, to re-establish soil fertility in low-input and no-input agriculture. In our crop planning experiments, we simulate a typical crop sequence that alternates soybeans and maize. At the beginning of each year, the agent observes the N profile of the soil and decides which crop to plant and when to plant it within a time window between the beginning of April and mid-June, discretized weekly. Its goal is to maximize profitability. No other operation, e.g., fertilisation or tillage, is performed.

We train our agents in an environment using weather data from Rocksprings between 1980 to 1998 randomly shuffled by year. Similarly, to Section 6.1, we denote this setting with *train*. To study the generalization of policies, we use three test environments using fixed sequences of weather: from Rock Springs between 1999 and 2016 (*test-time*), from New Holland between 1980 to 1998 (*test-location*), and from New Holland between 1980 to 2015 (*test-time-loc-horizon*). We study five non-adaptive baselines. The first four are standard fixed sequences: only soybeans (S), only maize (M), a sequence of soybeans-maize rotations (SM), and a sequence of soybeans-soybeans-maize (SSM). All of these fix the planting day to the value recommended in the cycles documentation. Here, we only give results for S and SSM which dominate the other fixed-sequence baselines. The last baseline is NONADAPTIVE, i.e., a policy trained with RL that only observes the complete crop sequence history as categorical variables, without feedback from the environment.

Table 3 shows the result of these experiments. For all methods, performances in the training scenario are higher than in other settings. The generalisation gap is wider for the test environment set in a new location and for the longer time horizon. However, the gap is smaller than 20% in all cases.

Interestingly, NONADAPTIVE agents achieve higher performance than RL ones. We conjecture this is due to the fact that the soil N profile disregards variables that greatly impact the dynamics, e.g., soybeans crop residue in the root zone and above ground. On the contrary, crop rotation history contains this information and, indirectly, captures the soil N content since we use a fixed soil initial condition and crops are the main driving force behind N dynamics in this scenario. Further analysis of this phenomenon are left for future work. The fixed baselines (S and SSM) are outperformed in almost all test modes by the best trained agents. Figure 5 shows the deployment of the deterministic policies learnt by RL and NONADAPTIVE agents in the *test-time* environment. The agents learn to alternate between soybeans and infrequent planting of maize, due to the poor yield obtained by

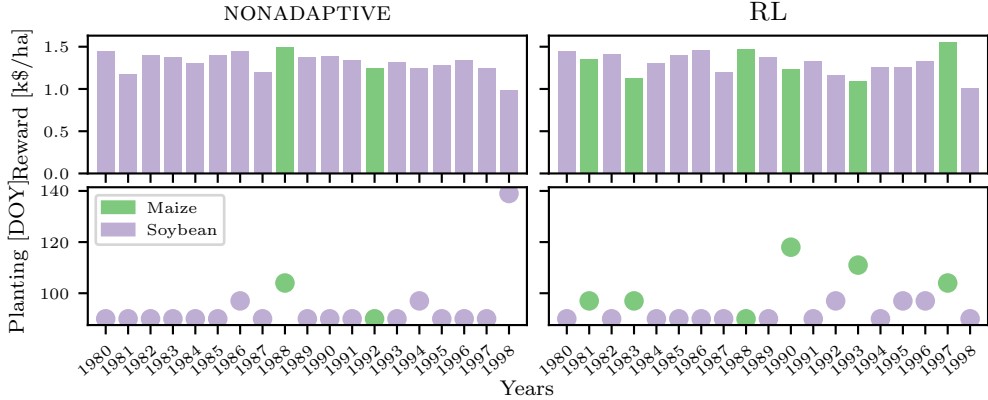

Figure 5: Crop planning policies for non-adaptive and RL agents. Top: yearly rewards in k$ per hectare, color-coded according to the crop planted. Bottom: planting date for the chosen crop.

consecutive maize cultivation without the addition of fertilizer. Furthermore, the RL agent learns different planting day patterns for the two crops. The soybean planting day learnt by the RL agents is usually around the beginning of April, which induces a higher yield thanks to the longer vegetative stage. Early planting date can also cause crop failure due to freeze at the beginning of the planting season. However, this was not observed in any of the simulations. This could be due to a cold tolerance and damage parameter that is too low in Cycles. Appendix D shows that training under fixed weather mildly hinders generalization. Moreover, it presents similar results in a more complex environment that includes main crop rotations and winter cover crops.

# 7 Conclusion

We present CYCLESGYM, the first RL environment for crop management tasks that incorporates long time horizons, multiple crops, and a wide array of possible management practices. Our experiments show that RL can learn effective policies outperforming domain-informed baselines in long-term N fertilization and crop rotation simulated tasks. Moreover, the experiments point to immediate research directions to bring RL closer to deployment in practical crop management. For example, identifying suitable (and approximately Markovian) observations that we can reasonably obtain in practice and that result in high-performing policies is necessary. Moreover, further work should consider using CMDPs to model logistics and environmental constraints. More generally, we present a series of challenges that must be addressed to improve RL's performance in simulated data-driven agriculture.

The most significant practical barrier for *direct* real-world applications are the substantial differences between simulation and reality that exist in crop management. Nonetheless, we believe our promising results in RL for simulated agricultural management together with advancements in calibration methods and Sim2Real algorithms represent a promising direction for sustainable smart agriculture.

# 8 Acknowledgements and Disclosure of Funding

The work of Matteo Turchetta and Scott Sussex was supported by the Swiss National Science Foundation under NCCR Automation, grant agreement 51NF40 180545. The work of Amanda Burton and Juan Herrera was supported by the Swiss Federal Office for Agriculture (FOAG) and *swiss granum*. The work of Ioannis Athanasiadis was partially supported by the European Union's Horizon Europe research and innovation programme, grant agreement 101070496 (Smart Droplets).

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
