# A  Discussion on Societal Impacts

This paper presents a library for creating RL training environments in the field of crop management. We think it is extremely unlikely that the release of this library would result in a negative societal impact. There could be negative impacts if decision-makers learn policies in our environments and directly apply them to high-stakes real-world settings without thorough evaluation. This is because, when transferring from simulated to real settings, there is a gap in performance due to the simulator not being a perfect world model. We make this clear when presenting our experiments, stating that performance should be considered only in the context of the environment and not be used to make strong agronomic claims. This is not a new risk; it has already been studied in the robotics literature where simulators are used frequently for training [29]. Further research on how to successfully transfer policies learnt in CYCLESGYM to real-world applications is a high priority. By providing an accessible toolkit for RL researchers to study some of the questions necessary for learning adaptive policies in crop management, we think it is much more likely that CYCLESGYM will have a positive societal impact by promoting long-term sustainable and optimal practices in agriculture.

# B  Experiment details

Here, we describe the experimental setup, including details about the crops and the locations of the simulations and the training of the algorithm. The results of such experiments are either in Section 6 or Appendix D.

**Crops**. In the US, maize and soybeans are grouped by time to maturity; relative maturity (RM) for maize and maturity group (MG) for soybean. Broadly, maturities are based on the relative thermal time required for the plant to complete its life cycle. In fertilization experiments, we use maize provided by Cycles in the *ContinuousCorn* simulations which use an RM 90 maize. For the crop planning simulations, we use an RM 100 maize and MG 3 soybean to better match the crop to longer local growing conditions in Rock Springs (central) and New Holland (southeast), Pennsylvania. These maturities are widely used by producers in these areas and are recommended by [16].

**Hyperparameters**. Fertilization and crop planning experiments use almost identical hyperparameters for training. These are the default hyperparameters for PPO policies in Stable-Baselines3 version 1.5.0 and they are used for both RL and NONADAPTIVE policy training. These settings include using a 2-layer multi-layer perceptron with width 64 and tanh activations. The only difference to the default hyperparameters is that in crop planning we update the policy every 80 environment steps instead of 2048. During training, observations and rewards are normalized using a running average during training. Crop planning agents obtain experience by interacting with 8 instances of the environment in parallel, while for fertilization experiments, agents interact with a single instance of the environment at a time.

**Other management practices**. For fertilization experiments we only vary N application in terms of amounts and timing. Likewise for crop planning we only vary the crop planted and date of planting. For both settings all other management variables are fixed and described in more details in Appendices B.1 and B.2. In both cases, we harvest on the first day the crop reaches physiological maturity, which is achieved by entering $-999$ for the HARVEST_TIMING variable in Cycles.

**Weather**. For training and evaluation, we consider both *fixed* and *random* weather. Fixed weather with start year $y_0$ and horizon $h$ gives a deterministic environment that always uses the $h$ years of weather starting from $y_0$ for the specified location. Random weather with start year $y_0$, end year $y_1$ and horizon $h$ gives a stochastic environment that uses weather where the first year $y$ is sampled uniformly from $[y_0, y_1 - h]$. The weather of the environment is then the $h$ consecutive years, in order, starting from $y$. We provide separate experiments for crop planning and fertilization experiments when training on fixed and random weather settings. Results where training was done in a fixed weather environment are marked "fw". The CYCLESGYM library more generally allows for various types of random weather sampling or the use of a generative weather model.

Below, we describe the experimental setup elements that are specific to either the fertilization or crop planning experiments.

Table 4: The observations for the RL environment used in N fertilizer experiments. Many explanations are based upon or taken from the Cycles user guide (https://psumodeling.github.io/Cycles/). The user guide also specifies measurement units of all soil and weather observations.

| Observation Name | Explanation |
|---|---|
| PP | total precipitation for the previous day |
| TX | max temperature for the previous day |
| TN | min temperature for the previous day |
| SOLAR | level of solar radiation for the previous day |
| RHX | maximum relative humidity in the previous day |
| RHN | minimum relative humidity in the previous day |
| STAGE | stage in the plant life cycle |
| CUM. BIOMASS | Cumulative plant biomass |
| N STRESS | daily N stress value of the crop |
| WATER STRESS | daily water stress value of the crop |
| ORG SOIL N | the sum of microbial biomass N and stabilized soil organic N pools |
| PROF SOIL NO3 | soil profile nitrate-N |
| PROF SOIL NO4 | soil profile ammonium-N |
| Y | years left in simulation (excluded for 1 year time horizon) |
| DOY | days since January 1st of that year |

### B.1 Fertilization

We use CYCLESGYM to construct a custom observation space for the RL approach. Table 4 specifies the observations the agent receives. Future work should consider what observations are most critical for learning and what observations can be effectively estimated with different kinds of sensors.

For NONADAPTIVE, we train a PPO policy with identical hyperparameters and training time but in an environment that doesn't include observations that allow the policy to adapt to feedback from the system. The observations are just 'DOY', 'Y', and 'N TO DATE'. 'N TO DATE' is the cumulative N applied so far in the episode. Because PPO learns stochastic policies, NONADAPTIVE policies were more successful when given access to this summary statistic for actions the agent had taken so far.

Every RL and NONADAPTIVE policy is trained for 3000 environment years. This means that shorter time horizons train for more episodes. Regardless of the training setup, we evaluate on the random weather setting. When evaluating trained policies on *test-time*, *test-location* and *test-horizon* we use 20 repetitions. We report the performance on these generalization tasks for the final policy obtained at the end of training. Static baseline (ZERON, PA, and CYCLES) performance is computed by averaging across 100 repetitions on all training and generalization tasks.

In all experiments maize is planted on the 110$^{th}$ day of the year. N applications for all methods consist of 75% NH4 and 25% NO3 in liquid form. These settings come from the *ContinuousCorn* operation files provided with Cycles.

### B.2 Crop planning

For training and evaluating RL methods, we use CYCLESGYM to construct a custom observation space that includes soil profile information. Table 5 gives the soil profile information included in the observations. For NONADAPTIVE we train a PPO policy with identical hyperparameters on an environment containing only a history of which crop the agent already planted in each previous year.

Every RL and NONADAPTIVE policy is trained for $170k$ environment timesteps, i.e., years (since each episode consists of a cycle of 18 years, this corresponds to training for approximately $9.5k$ episodes). All evaluations are done in environments with fixed weather. All generalisation metrics are evaluated regularly during policy training and for each repeat in our results we report the performance of the policy that has the best *training* performance on each of the generalization tasks.

Table 5: Soil profile observations for the RL environment used in crop planning experiments.

| Observation Name | Explanation |
|---|---|
| ORG SOIL N | the sum of microbial biomass N and stabilized soil organic N pools |
| PROF SOIL NO3 | soil profile nitrate-N |
| PROF SOIL NO4 | soil profile ammonium-N |
| MINERALIZATION | gross N mineralization due to decomposing organic substrates |
| IMMOBILIZATION | gross N immobilization in microbial biomass or stabilized soil organic N pools |
| NET MINERALIZ | net N mineralization |
| NH4 NITRIFICAT | nitrification of ammonium |
| N2O FROM NITRIF | nitrous oxide emissions from nitrification |
| NH3 VOLATILIZ | ammonia volatilization |
| NO3 DENITRIF | denitrification of nitrate |
| N2O FROM DENIT | nitrous oxide emissions from denitrification |

Table 6: The reward in k$ per hectare of agents on the *test-time* setup over different time horizons. We report the mean return over repeated runs, computed identically to as in Table 1. Standard error estimates are all below 0.01 so are not reported.

| | Horizon (years) | | |
|---|---|---|---|
| **Method** | **1** | **2** | **5** |
| **RL** | 2.07 | 2.04 | 2.03 |
| NONADAPTIVE | 2.04 | 2.00 | 1.98 |
| CYCLES | 1.94 | 1.69 | 1.60 |
| **PA** | 1.89 | 1.70 | 1.61 |
| ZERON | 0.78 | 0.50 | 0.42 |
| **RL-fw** | 2.09 | 2.08 | 2.07 |
| NONADAPTIVE-**fw** | 2.09 | 2.03 | 1.98 |

## C   Compute Resources

All experiments are performed on a compute cluster. For fertilization experiments, each repeat is performed using several CPUs and utilizing a max memory around 8GB. Experiments denoted RL also use a single NVIDIA GeForce RTX 2080 Ti or NVIDIA GeForce GTX 1080 for policy training in each repeat. A single repeat (training and evaluation) of RL for fertilization experiments takes approximately $8y$ hours where $y$ is the time horizon in years of the environment. A single repeat (training and evaluation) of RL for crop planning experiments takes approximately 48 hours.

## D   Additional Results

### D.1   Fertilization

In Table 6, we give the results of the *test-time* evaluation. Performance is similar to the *train* setting, though the gap to CYCLES is smaller suggesting a slight performance decrease when generalizing out of sample. Surprisingly, fixed weather RL has marginally improved performance in generalizing across time compared to RL trained on random weather. We think this is an artifact of comparing over a single set of test years. If we were to train and test with a wider array of locations and years, we would expect random weather to generalize better than fixed weather training, since the training objective for random weather is better aligned with the task of generalising across conditions.

In Table 7, we give the results for the *test-location* evaluation. Performance is reduced compared to *train* and *test-time*, however, fixed baselines are still outperformed. Some individual runs are outperformed by the CYCLES baseline. We find that fixed weather RL generalizes very poorly in a 1 year time horizon but performs comparably to and sometimes better than random weather RL over longer time horizons.

Table 7: The reward in k$ per hectare of agents on the *test-location* setup over different time horizons. We report the mean return over repeated runs and the standard error estimate, computed identically to as in Table 1.

| Method | Horizon (years) | | |
|---|---|---|---|
| | **1** | **2** | **5** |
| **RL** | 1.77 (0.02) | 1.78 (0.01) | 1.76 (0.01) |
| **NONADAPTIVE** | 1.75 (0.02) | 1.75 (0.02) | 1.77 0.01) |
| **CYCLES** | 1.74 (0.02) | 1.62 (0.02) | 1.55 (0.04) |
| **PA** | 1.71 (0.02) | 1.56 (0.03) | 1.51 (0.03) |
| **ZERON** | 0.72 (0.01) | 0.47 (0.01) | 0.41 (0.01) |
| **RL-fw** | 1.65 (0.00) | 1.79 (0.01) | 1.83 (0.00) |
| **NONADAPTIVE-fw** | 1.64 (0.00) | 1.73 (0.01) | 1.76 (0.00) |

Table 8: Results of the RL experiments in a crop planning scenario with winter cover crops. We provide the average yearly profit of baselines and RL agents in different environments in k$/(year·ha). The results are reported as mean (min, max) over 5 training runs with different random seeds.

| Method | *train* | *test-time* | *test-location* | *test-time-loc-horizon* |
|---|---|---|---|---|
| **SFSFMF** | 1.24 | 1.31 | 1.26 | 1.24 |
| **SFSF** | 1.27 | 1.31 | 1.22 | 1.22 |
| **SRSRMR** | 1.20 | 1.24 | 1.17 | 1.16 |
| **SRSR** | 1.27 | 1.31 | 1.22 | 1.24 |
| **SPSPMP** | 1.20 | 1.26 | 1.19 | 1.19 |
| **SPSP** | 1.27 | 1.31 | 1.23 | 1.24 |
| **RL** | 1.35 (1.35,1.35) | 1.30 (1.26,1.32) | 1.17 (1.09,1.21) | 1.09 (0.73,1.23) |
| **NONADAPTIVE** | 1.36 (1.35,1.37) | 1.33 (1.32,1.33) | 1.25 (1.23,1.26) | 1.01 (0.93,1.11) |

We provide training curves for RL on time horizons of 1, 2 and 5 years in Figure 6. This shows the performance in *train* over time as policies are trained. RL tends to reach higher performance with slightly fewer samples than NONADAPTIVE.

Figure 7 provides further examples of policy rollouts like the one given in Figure 3, again in the *test-time* setting. We visualize a separate 5-year RL policy, again using a deterministic version of the learned policy. We similarly visualize a stochastic version of the same policy that samples actions from the policy in the same way the PPO agent would during training. The stochastic versions tend to fertilize more often and are more likely to fertilize outside of the growing season. We also visualize a stochastic NONADAPTIVE policy. We do not visualize a deterministic version because we found that the deterministic versions of NONADAPTIVE policies are often degenerate. This is likely because the observation space for NONADAPTIVE agents contains significant partial observability. With significant partial observability in an RL environment, the optimal policy is not necessarily deterministic, meaning PPO may have converged towards a stochastic policy. We find that trained NONADAPTIVE policies are more likely to take unreasonable actions (such as occasionally fertilizing outside the growing season) compared to RL policies. This may partly explain the better comparative performance of RL agents in case there are no fixed costs.

## D.2 Crop planning

We provide the training curves for NONADAPTIVE and RL in Figure 8. NONADAPTIVE reaches higher performance in fewer samples but both converge to similar performance. On *test-time-loc-horizon* training curves are much less monotonic suggesting this is likely the most difficult generalization setting.

### D.2.1 Winter cover crops

In this section, we perform crop planning experiments that use both main crops (corn and soybeans) during the main growing season (spring and summer) and cover crops for the remaining part of the year (fall and winter). Cover crops are crops planted in periods when the soil might otherwise

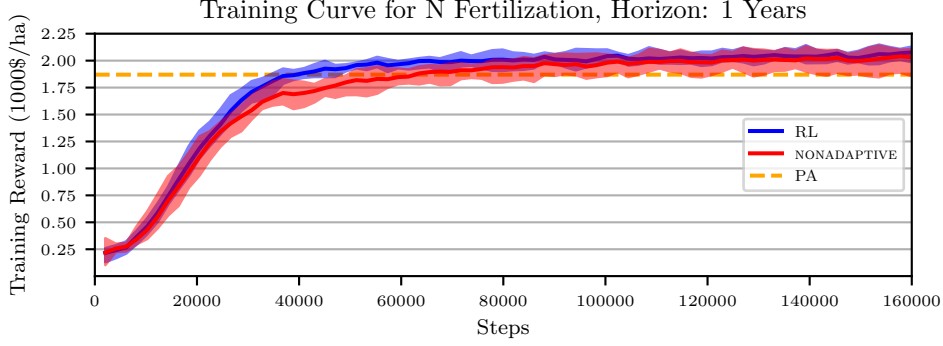

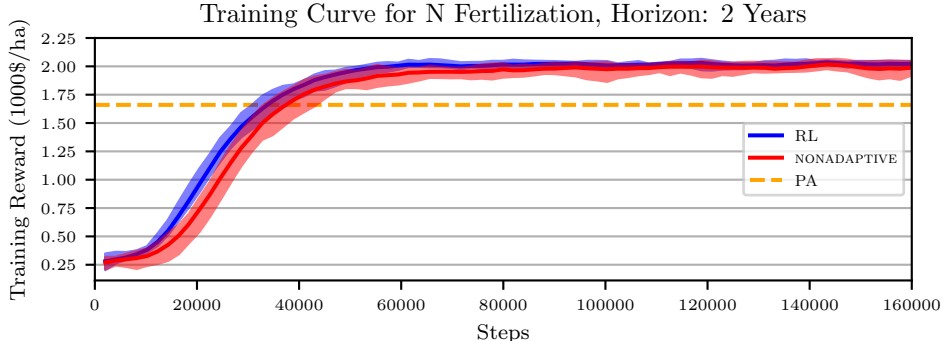

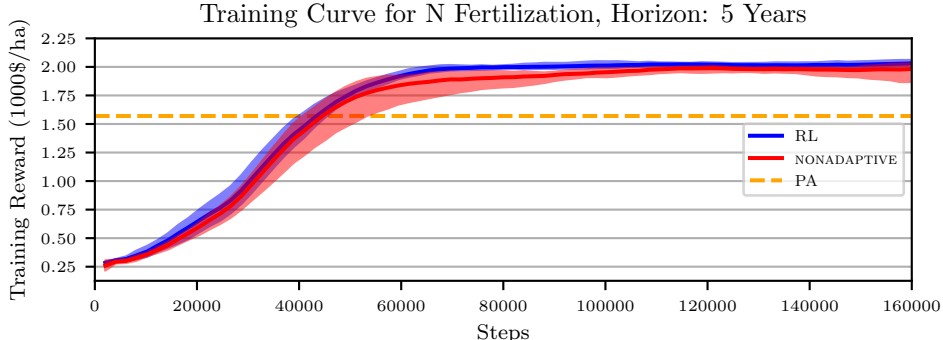

Figure 6: The performance of 1, 2 and 5 year N policies on *train* at various points during training. For each algorithm there are 5 runs. The bold line shows the mean performance across runs. The upper and lower parts of the filled regions show the max and min across runs respectively.

be fallow and are commonly used in agriculture to protect the soil from erosion and from loss of nutrients through leaching and runoff [17]. As winter cover crops, we consider ryegrass and winter peas, as they are widely used as cover crops in the region of the case study considered (Pennsylvania).

The action space in this scenario extends the one presented in the main paper. In particular, the varieties and planting dates for the main crops are those specified in Section 6.2. For the fall and winter period, the agents can decide to plant either winter peas (P), ryegrass (R), or keep the field fallow (F). In case they decide to plant a cover crop, the agents must also select a planting date between DOY 250 (first week of September) and DOY 306 in weekly increments. Since the action space considers a winter fallow period, the decisions considered in this experiment are a strict superset of those considered previously (see Section 6.2). Table 8 shows the results of the experiments in this environment. They differ slightly from those reported in Table 3, with small improvements achieved by the NONADAPTIVE strategy in the first three scenarios compared to the environment that does

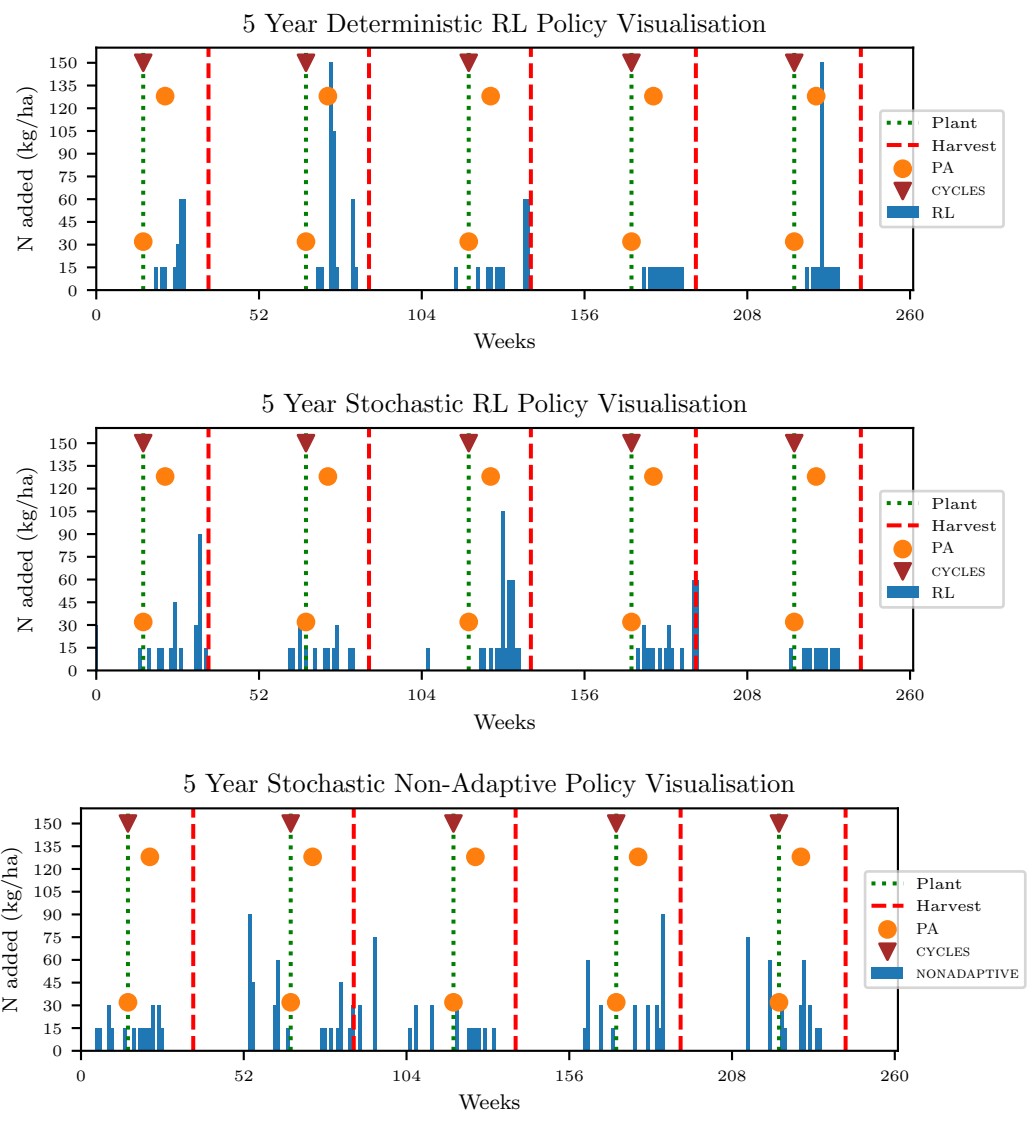

Figure 7: Visualizations of some 5 year policies. Top is a deterministic RL policy. Middle is a stochastic RL policy. Bottom is a stochastic NONADAPTIVE policy. Plots are over potentially different weather conditions.

not use winter cover crops. The same finding obtained in Section 6.2 is observed here, with the NONADAPTIVE outperforming RL in all scenarios.

# E   Code accessibility

At the time of the submission, the code is available as a *public* repository at `https://github.com/kora-labs/cyclesgym`.

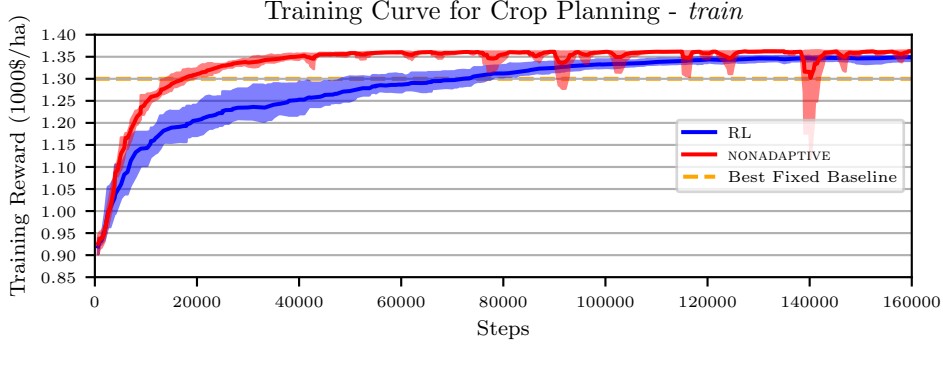

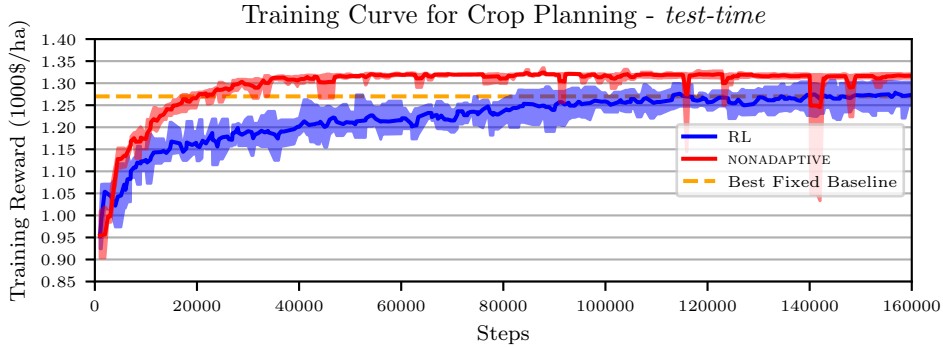

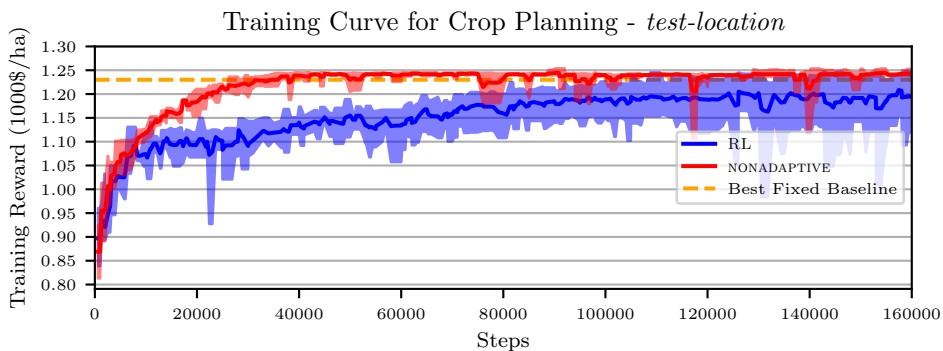

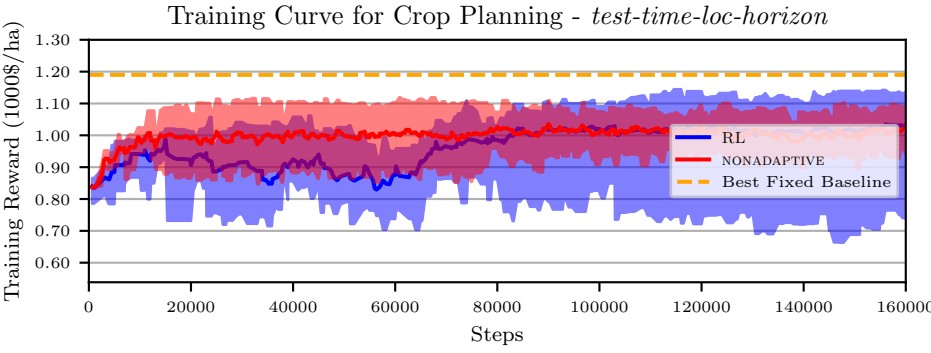

Figure 8: The performance of crop planning policies on *train*, *test-time*, *test-location* and *test-time-loc-horizon* at various points during training. For each algorithm there are 5 runs. The bold line shows the mean performance across runs. The upper and lower parts of the filled regions show the max and min across runs respectively.