# OpenReview forum: "Learning Long-Term Crop Management Strategies with CyclesGym"
_NeurIPS.cc/2022/Track/Datasets_and_Benchmarks — NeurIPS 2022 Datasets and Benchmarks _

### Official Review · Reviewer_MEYG · 2022-07-10
**RL Environment for Crop Modeling**

**Rating:** 5
**Confidence:** 5
**Clarity:** The text is well written.

**Strengths:**

Improving crop forecasting and understanding of the impact of crop management decisions is an important topic.  The environment seems to be well architected to allow for relatively easy simulation.

Analysis is performed looking at a very long time-horizon which demonstrates the multi-year capacity of the framework.

Results are given with error bars over multiple runs.

As a new RL environment, this appears to be well architected.

**Weaknesses:**

The idea of creating an RL environment is not new (it was done by many others as the authors cite), so the main contribution is around extending this to use a simulator which allows for multi-year modeling.  While including multi-year modeling is very important from an agronomic standpoint, I don't know that using a different simulator makes a meaningful enough contribution within the RL space.  The authors conduct several experiments to show that a policy is learned over many years, but the policy isn't meaningful (or significantly better).

A fundamental approach with all of these simulators is that they are woefully uncalibrated to the real world.  Even at the end of the season (when all of the actual information for that season is input), they can easily be off 30%.  Until that is addressed, it's unclear that these simulators have any real value.  The authors address that there is a sim-to-real gap that they are not addressing.  However, without speaking to just how massive this gap is gives the appearance that this is just the standard RL challenge and that with some effort, can be overcome to make real world impact.  Unfortunately, the situation is far more complicated than that and until significant work is done, these agronomic simulators are no "real" than training RL to play a video game.  Without this impact, it's unclear that this simulator is needed to address any of the RL research questions on page 5 (i.e. those questions can be explored with other environments).

For example, in the analysis around N fertilization, it is unclear if the learned policy is realistic.  Farmers will only spray so many times (it takes effort, equipment must be moved around) a season.  Additionally, the growing stage of the crop dictates how the N is applied, and therefore the cost, type of N, and type of equipment required.  While these seem like minor details, they are fundamental considerations which the farmer addresses.  Therefore if the policy does not take this into account, it is again reduced to a toy problem with minimal real world usability.

Nitrogen application cannot be optimized without weather and water considerations.  Doing N alone is fine to demonstrate the code runs properly (i.e. it serves as a "unit test" in engineering language), but it's not a realistic result.

To demonstrate the value of the simulator, multiple variables need to be allowed to vary, not held fixed.  Yes single variable experiments are useful to ensure it has been coded properly, but to demonstrate it has value to the community, realistic experiments must be conducted.  Similarly, a simple baseline approach is fine to show that it works, but the point of this simulator (even from just the RL standpoint) is to address some of questions the authors articulate- I think the authors need to take a step (not solve it, just one step above baseline) in one of those directions again to show what can be done with this framework.

**Additional Feedback:**

This is a worthwhile line of work.  However, I think it needs to be clearly stated what the current limitations are.  Right now, these approaches are only useful to the RL community and do not have any real world contribution to agriculture because the underlying crop simulators are so poor and limited.  The sim-to-real gap here is significantly larger than it is in other domains, which is saying something.  Therefore I think it is incumbent on the authors to state these significant limitations and discuss how they will be overcome.  They don't have to be solved in this work, but it's important to realistically state where this work stands and what must be done for it to reach a state of impact.  Otherwise it needs to be cast as purely a toy problem and all of the discussion around world hunger, nitrogen supply chain, etc. need to be removed because it can't realistically be addressed with this approach.

**Correctness:**

See concerns above around the constraints of the experiments.  I don't doubt the experiments were run correctly, but that they are not sufficient to demonstrate value and stability.

**Documentation:**

This is a generation framework.  Documentation around the author's contribution appears to be clear.  It may be helpful to supplement the Cycles documentation with information that makes the underlying simulator clearer to a non crop-science audience.

**Relation To Prior Work:**

Other crop-RL work are mentioned, however, I believe more could be said here.  Most are simply a statement of what that work did, although a few mention limitations around those approaches.  The specific task (irrigation, nitrogen application, etc.), is largely irrelevant because of the lack of real-world impact (also it's relatively trivial to change which task you want to optimize for). Instead, I believe the authors should address what types of RL analysis cannot be provided with those other approaches, and what benefits/limitations exist with the other crop simulators.  The benefit of Cycles is that it's multi-year, but how much is traded in terms of speed, ease of extension, number of crops, etc.?  That type of comparison needs to be made clearer.  I would suggest significantly extending this section and possibly including a table which emphasizes these points.

**Summary And Contributions:**

The authors construct an RL environment to enable evaluation of multi-year management decisions on crop performance.  This environment is based on the Cycles agronomic simulator.

A key challenge all of these agronomic simulator-RL approaches have is that the simulators are not reliable.  That's not the fault of these authors, however, because the simulators are so untrustworthy in many cases, it reduces a problem which appears to have very meaningful societal impact to a toy problem, useful only for studying RL techniques.  There is still some value to the RL community there (it is another environment to test algorithms there), however, the impact of the work is much less than it would be if these simulators were better.  The sim-to-real domain gap exists everywhere for RL, but it is particularly bad with many of these crop simulators.  Either the real-world impact is limited, or the authors must address how this gap is to be addressed in order to bring these results into the real world.  I appreciate the need to take small steps in solving this problem, but I think the gap and therefore the impact needs to be realistically addressed.  Since no agronomic insight can be drawn from these models due to the size of the domain gap, the only contribution is to the RL community, and it's not clear that this environment opens up new avenues of exploration.

---

> ### Author Response · Authors · 2022-08-10
> **Reply to R MEYG**
>
> We would like to thank the reviewer for raising valid and important points.  Please find our answers below.
>
> **Multi-year benefits**
> Please, see our “Multi-year benefits” common reply at the top for a discussion of the benefits and “Comparison to other crop RL environments” for a discussion on why our new environment is needed.
>
> **Relevance of crop growth models**
> CGMs are widely used as a support tool across many organisations providing agronomic recommendations and policies around the world (e.g., the CGIAR organization [1] and FAO [2]).
>
> Due to the need to calibrate them, CGMs are not yet popular for direct producer use. However, when calibrated, they achieve higher accuracies than those outlined by the reviewer [3]. Therefore, researchers use them to explore questions that are difficult or costly to answer with field experiments, or as a preliminary step before deploying field studies. As a result, we believe cyclesgym represents a relevant benchmark for RL and the Ag community, provided that its Cycles backend must be calibrated to address a real-world case study. Calibration methods for CGMs are outside the scope of this work but are currently explored by the authors and by the community at large.
>
> In conclusion, we agree that the reality gap is a particularly sensitive problem with CGMs but not an insurmountable one. In the original version of the paper, we had an extensive discussion of this limitation in Appendix A. In the revised version, we also discuss this in the research questions section and in the conclusion of the revised version.
>
> [1] Kruseman, Gideon, et al. "CGIAR modeling approaches for resource‐constrained scenarios: II. Models for analyzing socioeconomic factors to improve policy recommendations." Crop Science 60.2 (2020): 568-581.
>
> [2] Masseroni, Daniele, et al. "Comparing different FAO approaches for assessing irrigation needs and scheduling: application on a maize field in Mediterranean area." EGU General Assembly Conference Abstracts. 2021.
>
> [3] Ahmed, Mukhtar, et al. "Models calibration and evaluation." Systems modeling. Springer, Singapore, 2020. 151-178.
>
>
> **Realistic scenarios**
> Please see our common reply at the top.
>
> **Water and weather considerations**
> In our fertilization experiments, crops are rainfed and the agents do take into account weather information that is provided to them in the observation.
>
> **Multi-variable experiments**
> The Cyclesgym library is intentionally open-ended, allowing practitioners to craft their own environments relevant to their real-world tasks. Since the space of possible environments is too large for us to explore in this work, we demonstrate experiments on simple settings that consider a single management variable (planting and N management).
>
> **Why cycles?**
> Please see our common reply at the top.

---

### Official Review · Reviewer_zvAw · 2022-07-24
**CyclesGym Review**

**Rating:** 7
**Confidence:** 4

**Strengths:**

[S1] The benchmark includes two important problems in crop management, fertilization and crop rotation, as well as long-term experiments. Compared to prior environments suggested for RL application to crop management, these allow for a more thorough analysis.

[S2] The baselines for experiments chosen are appropriate and allow for strong evaluation of new RL methods.

**Weaknesses:**

[W1] While Cycles appears to be a reasonable choice of underlying model, it is under active development and supported by research, it would be helpful if some justification for the choice was given. Under related work discussing CGMs it appears the choice was arbitrary with other models potentially a better choice as they allow for ground water models and CO2 concentration modeling. Are these other models less suited to modelling long-term effects than Cycles?


**Additional Feedback:**

[A1] Given the large number of CGMs, could the authors speak to the flexibility of the benchmark in potentially operating these other models?

**Clarity:**

[CL1] The paper is very well written with key points easy to understand. However, it may be pertinent to include the duration of the RL training in the main paper as well. The value of sim-to-real transfer is somewhat different in the case that algorithms operate over a realistic time scale or over thousands of simulated years.


**Correctness:**

The experiments highlight issues in applying RL to CGM models with appropriate baselines for evaluation.

[C1] It would be helpful to include the standard deviation along with the mean performances. This is useful in gauging the stability of the learned policies.

**Documentation:**

[D1] The code for the benchmark is not available for review, even with the change in access noted. To retrieve the code for github authorization, a gmail account is provided. That gmail account also requires authentication.

**Ethics:**

The authors appropriately caution the application of models learned from the benchmark directly being applied in the real-world without further research.

**Relation To Prior Work:**

[R1] While one of the claimed benefits of the proposed benchmark is multi-year capability, it's not clear how this capability was lacking from prior works. Is there some inclusion which is different from setting the underlying simulator's time frame?

**Summary And Contributions:**

The proposed benchmark improves on previous RL environments for an important domain with widespread consequences. The provided experiments and evaluation allow the authors to identify the more pressing issues of RL algorithm design when applied to crop management.

---

> ### Author Response · Authors · 2022-08-10
> **Reply to R zvAw**
>
> We thank the reviewer for the insightful feedback. Please find our comments below.
>
> **Why Cycles?**
> Please see our common reply at the top.
>
> **Experiments**
> We have made some updates to the reporting of results including performing more runs and computing the standard deviation of the estimate of mean performance. See the common response for more details.
>
> **Multi-year capability**
> Please see our common reply "Comparison to other crop RL envs" at the top.
>
> **Code accessibility**
> Please see our common reply at the top.
>
> **Support for other CGMs**
> Please see our common reply at the top.

---

> > ### Comment · Reviewer_zvAw · 2022-08-28
> > **Reply to Rebuttal**
> >
> > Thank you for addressing my concerns. I have no further questions.

---

### Official Review · Reviewer_yZVT · 2022-07-25

**Rating:** 6
**Confidence:** 4
**Correctness:** Yes.
**Clarity:** Yes.

**Strengths:**

1. The paper is well written and motivated. Comprehensive review is provided for the related work in Machine learning in agriculture, Crop growth models and Crop management with RL.

2. The targeted problem of RL applications in crop management/agriculture is understudied and has great potential for societal impact.

3. The discussion related to the application of RL in crop management/agriculture is interesting and enlightening, such as sec 4.2, which could be helpful for researchers in both fields.

4. The experiments are interesting, which encompass relevant crop management strategies under some realistic senarios (partial observability, temporal/location shift between train/test data)

**Weaknesses:**

1. Limited contributions in datasets/environments. CyclesGym seems to be a direct wrapper around the multi-year, multi-crop CGM Cycles, the contributions to data/environment and crop growth modeling are limited.

2. Experiments could be stronger.
- If datasets/environments aren't the focus, as a benchmark paper on RL application in agriculture, this work only evaluates some variants of a single RL algorithm, PPO. Adding more RL or planning/optimization algorithms would make the paper stronger, such as genetic algorithms [1] and SAC [2] . Meta-RL [3] would also help since CyclesGym can in principle create many distinct environments/tasks. Also, it would be nice to provide empirical studies and implementations of the RL research questions discussed in 4.2. More fine-grained control in action space and time scale as in [1] would be interesting to see, too.

- Moreover, I think it would be very helpful to benchmark an RL oracle (i.e. the theoretical upper bound) on each set of the experiments. An important point of benchmark is to create a task challenging enough to allow for further algorithmic innovation. Therefore even though RL algorithms like PPO has shown marginal improvement over common strategies, it's unclear how much room of improvement is left for further development. Such experiments would be valuable to the machine learning community.

3. Better accessibility to code and datasets. The current access to the cyclesgym repository is hindered by email/device verification, making it hard to assess the implementation quality and reproducibility of the work.

[1]: Xiaoyan Cao, Yao Yao, Lanqing Li, Wanpeng Zhang, Zhicheng An, Zhong Zhang, Shihui Guo, Li Xiao, Xiaoyu Cao, and Dijun Luo. igrow: A smart agriculture solution to autonomous greenhouse control.Association for the Advancement of Artificial Intelligence (AAAI), 2022.

[2]: Tuomas Haarnoja, Aurick Zhou, Pieter Abbeel, and Sergey Levine.  Soft actor-critic:  Off-policy maximum entropy deep reinforcement learning with a stochastic actor. In International conference on machine learning, pages 1861–1870. PMLR, 2018.

[3]: Duan, Yan, et al. "Rl $^ 2$: Fast reinforcement learning via slow reinforcement learning." arXiv preprint arXiv:1611.02779 (2016).

**Additional Feedback:**

The authors might consider adding more to the RL research questions (sec 4.2) and experiments in agriculture:

1. Offline RL. Given the large gap of characteristic time scale between crop growth and RL training, it's impossible to train an RL model by online interaction with a real environment (even for sim-to-real paradigm, the trained model needs to perform some kind of online adaptation). Hence offline training seems to be a more practical setting. It would be interesting to see if CycleGym can provide testbed/benchmark for single-task offline RL [1] and multi-task offline RL algorithms [2][3], too.

2. Delayed and sparse reward. In principle, the crop management devision process takes actions on a weekly or even daily basis. However, when calculating profit, crop harvest/production can only take place rarely with time delay. It would be intersting to see if CycleGym has capacity for simulating such real-world scenarios and how the corresponding RL algorithms perform.

[1] Levine, Sergey, et al. "Offline reinforcement learning: Tutorial, review, and perspectives on open problems." arXiv preprint arXiv:2005.01643 (2020).

[2] Furuta, Hiroki, Yutaka Matsuo, and Shixiang Shane Gu. "Generalized decision transformer for offline hindsight information matching." arXiv preprint arXiv:2111.10364 (2021).

[3] Li, Lanqing, Rui Yang, and Dijun Luo. "Focal: Efficient fully-offline meta-reinforcement learning via distance metric learning and behavior regularization." arXiv preprint arXiv:2010.01112 (2020).

**Documentation:**

Yes.

**Relation To Prior Work:**

Yes.

**Summary And Contributions:**

This paper introduces CyclesGym, an RL environment based on the multi-year, multi-crop crop grow model (CGM) Cycles for open field agriculture. On CyclesGym, the authors benchmark two types of crop management practice: nitrogen fertilizer (N) application and crop rotation. They perform comparative studies of RL policies vs baselines representing current practices, which are evaluated with train/test splits in time, location, horizon.

The key contributions of this work include:

1. It is the first crop-management RL environment and benchmark that facilitate the learning of multi-year strategies with complex action spaces and multiple crops.

2. For RL researchers, this is a novel realistic benchmark that offers the possibility to investigate relevant issues arising in real-world RL while tackling a pressing societal problem.

3.  For agronomists and environmental scientists, CyclesGym allows users to leverage recent advances in RL to improve management practices in agricultural systems.

---

> ### Author Response · Authors · 2022-08-10
> **Reply to R yZVT**
>
> We thank the reviewer for the helpful feedback. Please find our comments below.
>
> **Environment contribution**
> Cycles was not developed for the ML community. Therefore, it is not suited to fit in an RL loop as it requires the user to pre-specify management decisions before running a simulation, rather than taking them interactively in response to weekly observations. Wrapping it in an RL environment requires a considerable engineering effort. Notice that the same wrapping approach is also adopted by gym-DSSAT [24], the other RL environment based on a complex CGM.
> Concerning the modelling contribution, Cycles is a complex software that implements the results of decades of research in CGMs and extending it is beyond the scope of our work.
>
> **Experiments**
>  - Other RL algorithms: We use PPO as it typically performs well in RL studies and it is robust. We experimented with deep Q learning and observed that its convergence is slower than PPO in the 1-year fertilization experiments. Since our main contribution is a new environment that can be used to create many scenarios, we did not explore this algorithmic comparison in detail.
>  - Meta-learning and Section 4.2: The challenges highlighted in sec 4.2 (including meta-learning) are relevant RL open questions that one can investigate with Cyclesgym. Solving them requires remarkable research effort that goes beyond the scope of this paper, which presents a new benchmark rather than new algorithms.
>  - Fine-grained actions and time: We ran experiments with fine-grained actions and there was no performance benefit while convergence was slower. Concerning the finer time resolution, there is not a significant performance benefit. However, it makes training slower.
>  - Oracle: To obtain an upper bound, we need to provably find the globally optimal policy, which we cannot do in RL except for the simplest problems. However, even if PPO were approaching the optimal value, there would still be room for improvement, e.g., sample efficiency, robustness to noisy observations, and more.
>
> **Code accessibility**
> Please see our common reply at the top.
>
> **Additional research questions**
> Offline RL: Due to the long turnover times of agriculture, we agree that offline RL is a promising method in this context. We will add this in the final revision.
>  - Sparse and delayed rewards: In our N experiments, agents are penalized every time they fertilize but rewarded only at harvest. Thus, the reward is sparse (once per year) and delayed. In the crop rotation experiments, agents are rewarded once per year but, since the time step is one year, the reward is dense and instantaneous.

---

> > ### Comment · Reviewer_yZVT · 2022-08-29
> > **Reviewer Response**
> >
> > Thank you for your reply. Overall I recognize the value of the problem this paper tackles and think the paper does show some interesting results. However, as the authors claimed that the major contribution of the paper is providing a new RL environment, I think the significance of the paper would be largely dependent on the technical/engineering challenges addressed by wrapping the Cycle CGM. Can the authors elaborate on this point?
> >
> > Also I agree that the sim-to-real gap pointed our by reviewer MEYG is a real and major concern, regarding the significance of the paper.
> >
> > Lastly, a minor point: you expressed agreement in adding offline RL discussion in the final revision, which seems to be absent in the current version.

---

> > > ### Author Response · Authors · 2022-08-29
> > > **Cyclesgym implementation**
> > >
> > > We would like to thank you for your feedback.
> > >
> > > Cycles is distributed as an executable and the only way to interact with it is by reading and writing unstructured text files. To wrap it in a Python environment, it is necessary to define managers that can parse/save all the different types of text files that exist within Cycles. Given these, one can write an environment around Cycles by reading the files necessary to compute the observations, passing the relevant variables to the agent, writing to the operation file the management practice recommended by the agent, reading the files for the reward computation, and repeating. On one hand, this kind of environment already requires some engineering to develop that involves handling corner cases, time consistency, and action feasibility. On the other hand, it would be slow, hard to modify, and not easily accessible. To address this, Cyclesgym implements multiple improvements. Below, we list the most important ones:
> > >
> > >  - **Careful file management**: This allows for parallel runs of the same environment (fundamental for RL training). This is also crucial for fast runs compared to a naive implementation as the files involved are large.
> > >  - **Fast multi-year simulations**: Simulations spanning multiple years are faster compared to a naive implementation because we leverage Cycles' reinit files to dump intermediate results and restart simulations from when they were interrupted.
> > > - **Lazy evaluations**: We make the environment considerably faster by calling Cycles as few times as possible and reusing the results of previous simulations.
> > >  - **Modularity**: Our architecture allows us to easily define new scenarios by combining different observations and rewards or by specifying generative models for the weather (something that does not come natively with Cycles).
> > >  - **Tests, tutorials, and documentation**: While these components do not directly impact the wrapping of Cycles, they are a crucial part of reliable and accessible software and require time and effort to develop.
> > >
> > >
> > > Concerning offline RL, we have added a short (due to the limited space) paragraph in Section 4.2.

---

### Official Review · Reviewer_GiWq · 2022-07-27
**Potential new benchmark for RL in crop rotation simulation**

**Rating:** 6
**Confidence:** 5
**Correctness:** 1) Mention on the number of replicate…
**Clarity:** 1) While the text is well written, it…

**Strengths:**

1) Application of an OpenAI gym to GCM could allow new research opportunities in agronomy.
2) Multiple-Year simulations and possibility of adding policies.
3) Usage of the OpenAI Gym allows future optimisations and new RL models to be tested.


**Weaknesses:**

1) Limited documentation for the benchmark and how the planted superficies (ha) are handled in the simulations.
2) CycleGym architecture should be more defined in section 5 with some examples.
3) Limited geographical locations (US, New Holland and Rock Springs ) and rotation for the simulation not representative of the whole World and agronomic practices.
4) The research question is not well defined in the introduction. There are mentions of the Ukraine war, Pest control, CO2 emission, and N pollution, while the subject of crop rotation and management strategies, central to this benchmark, are not mentioned and explained.
5) No real description of each year's weather patterns which can influence the results presented in the benchmark.
6) No mention of the OpenAI Gym in the document as being the RL engine (only in the Github)
7) The RL model policies used by the authors are  not well described (maybe a schematic /table view) in terms of the goal and rewards. This information is covered in part in section B, but should be in the main part of the manuscript.
8) New simulations and results in Table 5 (Annexe) (RL-fw and NONADAPTIVE-fw) not explained.
9) RL reward is based on profit (k$/y ha), however, raw yield should be used since corn/soybean prices also fluctuate on a yearly-basis.
10) The title of this manuscript is Long-Term Crop Management Strategies. However, this is not well documented in this manuscript. While the manuscript is oriented  in computer science, this aspect should be discussed in greater length.
11) Some of the Figure captions and Table descriptions needs more details.
12) Conclusion/discussion section is too short.


**Additional Feedback:**

Re-focus the manuscript on RL and its implication in digital twin in agriculture. Focus on only one problematic in agriculture (crop rotation or fertilizer). Had a greater discussion on the plus and minus of the dataset and benchmark environment provided and better explain some of the simulation (e.g. RL-1 vs RL) and the different result implications.

**Documentation:**

1) Availability is poor since the Github is not public
2) Documentation is not present and code samples is not too explicit.
3) Litterature cited is diverse and not centered on the problematic demonstrated (corn/soybean rotation)

**Ethics:**

No problem

**Relation To Prior Work:**

This is not well established by the authors. While the benefit of this benchmark could be helpful, how to interact and modify it to test other hypothesis is not clearly stated.

**Summary And Contributions:**

In this manuscript, Turchetta et al. propose a new environment for crop growth models (CGM) simulation, where agronomists and researchers can try different reinforcement learning (RL) or ML strategies. The simulator is, as described, a kind of wrapper around the software Cycle, one of many CGM that can be used for simulation of crop growth and crop rotation. This is not a novel subject, since GCM and Decision support systems (DSS) like CROPGRO (Hoogenborm et al. 1991), STICS, DSSAT, DNDC and HOLOS are used by different state agencies (e.g. USDA, INRAE) to predict outcomes for different crops during the growing season. One key limitation is often the real dataset (e.g. weather information, real crop yield, financial data) that needs to be used to confirm the validity of the different models. As such, different models are normally run concurrently since they all have some biais. And this is probably what is the biggest drawback of this study and proposed benchmark: 1) the support of only one CGM; 2) the use of unrealistic agronomic practices for their simulations (e.g. multiple N applications during a season, only some crop rotation); 3) looking at the code on the github, it is unclear how one could change the simulation parameters to fit its need (e.g. changing the weather, crop) using the provided documentation. Nevertheless, a more open system with easier configuration options (e.g­. Weather, Location, Crop) could provide an interesting benchmarks environment for RL in agriculture.

---

> ### Author Response · Authors · 2022-08-10
> **Reply to R GiWq**
>
> We thank the reviewer for the detailed and relevant feedback. Please find our replies to the main concerns below. Minor concerns will be addressed directly in the final revision.
>
> **Support for other CGMs**
> Please see our common reply at the top.
>
> **Realistic scenarios**
> Please see our common reply at the top.
>
> **Documentation**
> Please see our common reply at the top.
>
> **Clarity**
> We will address the concerns about the clarity and the focus of the text in the final revision, compatibly with the page limit.
>
> **Why Cycles?**
> Please see our common reply at the top.
>
> **Comparison to other crop RL environments**
> Please see our common reply at the top.
>
> **Code accessibility**
> Please see our common reply at the top.
>
> **Focus on a single application**
> Including two applications is meant to demonstrate Cycle’s flexibility. In particular, N fertilization is the application studied in closely related works [41, 62], and crop rotation is a new application that cannot be investigated in existing RL environments for agricultural management.
>
> **Reward definition**
> Using raw yield neglects the trade-off with the costs of inputs such as N fertilizer, which are also variable. In a situation where maize price is low and N price is high, maximizing raw yield may not be desirable. The modularity of our library allows for users to easily hand-design a reward function if they would like to focus on an objective besides profit.

---

> > ### Author Response · Authors · 2022-08-29
> > **Reply to R GiWq (crop rotation experiments)**
> >
> > We updated the main text and the supplementary material with a new experiment on crop planning. The new environment used includes both main crops (as the environment used already in the main paper) and additionally winter cover crops.

---

### Official Review · Reviewer_rNfn · 2022-07-27
**A Benchmark with Potential**

**Rating:** 7
**Confidence:** 3
**Correctness:** See the above section on weaknesses o…
**Clarity:** The paper is clear and understandable.

**Strengths:**


CyclesGym is the first benchmark to evaluate and train RL algorithms on multi-year, multi-crop agriculture. As such, the benchmark could potentially have positive societal impacts by fostering the development of RL algorithms which are better suited for applications to agriculture.

The benchmark is useful to the community studying applications of RL to agriculture. Current benchmarks focus on single-year agriculture.  CyclesGym allows for multi-year, multi-crop environments, where the decisions of previous years have long-lasting effects. For the agronomist, this environment could provide a more realistic simulation of the real world than current benchmarks.

**Weaknesses:**

The experiments in the paper are intended to show the utility of CyclesGym and how this environment is useful over existing environment. These experiments should have been conducted with more rigour.

Experiment 1 attempts to outline the benefits of CyclesGym compared to current benchmarks in terms of the multi-year feature of CyclesGym. PPO is trained for both a 5 year (PPO-5) and 1 year (PPO-1) period and then evaluated on a 5 year period. If the performance of PPO-5 was significantly higher than PPO-1, then this would provide evidence that CyclesGym is useful to study crop management over multiple years, or that the multi-year feature of CyclesGym results in a harder problem.

Unfortunately, the benefit of CyclesGym over current benchmarks in this experiment is unclear. In Table 1, mean performance with min/max as uncertainty in performance is reported. The conclusion is drawn that PPO-1's performance deteriorates with more years. Nevertheless this conclusion is unclear since the reported uncertainty overlaps in all cases.

Even if the paper reported confidence intervals, I am uncertain that PPO-5 would statistically significantly outperform PPO-1, bringing the utility of CyclesGym into question, at least with respect to the multi-year feature. With only 5 random seeds used, the estimated mean performance is most likely inaccurate. It could be the case that PPO-1 saw a number of low-performing runs but that with more runs, the mean performance of PPO-1 and PPO-5 would be nearly the same. This experiment would be improved by increasing the number of runs used and reporting statistical measures of confidence.

Furthermore, the results for a number of baselines do not include any uncertainty measurements (for both experiment 1 and 2). Were these baselines only run using a single seed? Are both these baselines and the experiments with CyclesGym deterministic such that a single experiment was needed? PPO in these experiments does not take into account the cost of supplies, operating equipment, etc. Do the baselines takes these factors into account? If so, this could give PPO an advantage.

In experiment 1, generalization between locations is performed by training with the weather of one location and testing with the weather of another location.  Is weather the only differences in locations in the benchmark? If so, why refer to this setting as *locations* and not *weather* or *climate*?  Location brings with it many connotations other than weather, e.g. soil quality, prices of supplies, air quality, average UV index, etc.


**Additional Feedback:**

*These did not affect the scoring of the paper*:

- The paper mentions that CMDPs are a more principled way of incorporating constraints than negative rewards, but does not give a full justification. I am not familiar with CMDPs, but a potential issue is that many existing algorithms designed for MDPs may not be compatible with CyclesGym. Is it possible to build constraints into the reward function as well, as perhaps a separate version of the benchmark?

- Line 138: The paper reads: "The action space contains all the decisions...", which makes it sound like a trajectory. Maybe state "The action space contains all the possible decisions..."

- Line 265: I believe the references should be to tables 5 and 6 not 2 and 3.

- Lines 81-82: a LED -> an LED

- Inline references read nicer when the authors names are used. E.g. "Sutton et al. found that ..." rather than numbers as used in the paper: "[30] found that...". I found myself getting confused while reading on what was an inline citation and what wasn't.

- It might be useful to have major documentation in text or markdown files as well. Some documentation I could only find in the included notebooks, but it might be easier to read this documentation if it is in a markdown file.


**Documentation:**

CyclesGym seems to lack documentation. I would expect each public class, function, etc. to have a docstring, but many of these docstrings are missing.


**Ethics:**

The paper addresses appropriate ethical concerns in the appendix.


**Relation To Prior Work:**

I do not have a strong background on the area of agriculture and RL, but from what I can tell, the paper places itself in the literature appropriately.


**Summary And Contributions:**

CyclesGym is a new benchmark environment for reinforcement learning (RL) applications to agriculture and farming.  The benchmark allows for different crops to be produced over a span of multiple years.  The environment can also be modified and customized to the needs of the experimenter with new crops, locations, etc.

---

> ### Author Response · Authors · 2022-08-10
> **Reply to R rNfn**
>
> We would like to thank the reviewer for the feedback concerning the experiments.  Please, find our answers below.
>
> **Multi-year benefits**
> Please, see our common reply at the top.
>
> **Uncertainty of Baselines**
> The uncertainty estimates in the tables for RL methods depend on the randomness in the testing environment (due to the weather), the policy, and the training process. Since the baselines are not trained, deterministic, and independent of the observations, their uncertainty is only attributed to the weather. In the revised version, we include uncertainty estimates for the baselines computed with Monte Carlo approximation (100 repeats) and apply the central limit theorem. The uncertainty in estimating the mean is small and we can say confidently that RL and nonadaptive have higher performance than the baselines.
>
>
> **Advantage of PPO**
> Please see our “realistic scenario” reply at the top.
>
> **Difference between locations**
> In experiment 1, the only difference between the locations is the weather. This is because Cyclesgym ships with the weather files for these two locations and a generic soil file, which is suitable for both of them. We could not find data on different prices of supply at such a refined level (both locations are in Pennsylvania).
>
> **Documentation**
> Please see our common reply at the top.
>
> **CMDPs**
> Our CMDP implementation uses the interface recommended in Ray et al.  "Benchmarking safe exploration in deep reinforcement learning.", 2019. Crucially, this interface preserves compatibility with standard MDPs implemented with an OpenAI gym interface. This means that standard RL algorithms for MDPs can be seamlessly applied to solve the unconstrained version of our environments without any modification. As a result, a separate version of the benchmark is not necessary.

---

> > ### Comment · Reviewer_rNfn · 2022-08-16
> > **Reply to Rebuttal**
> >
> > I thank the authors for the clarifying comments as well as for the additional experiments which have addressed the majority of my concerns. I have increased my rating accordingly.

---

### Comment · Reviewer_GiWq · 2022-07-27
**Unable to access the code, and above Github URL is wrong**

Good Morning,

I'm in the final review stage. Unfortunately, i'm unable to access the validity of the research since I'm unable to view either the code, or some of the RL results (besides the ones shared in the Annexes).

It is thus dificult to evaluate the work, Maybe the author should include a Colab notebook or similar with some examples?

* P.S. When I try the different email and credentials proposed in the thread or in the annexe, I'm always locked up with a device verification stage.

---

### Author Response · Authors · 2022-08-10
**Comments on common concerns**

We thank the reviewers for their time and effort.

We have updated the manuscript (new experiments and improved text) to incorporate the main points discussed here. We will include minor points in the camera-ready version. The heavily modified parts are highlighted in red in the revised document.

We have improved the documentation on Github and we will keep doing so during the response period.

**Multi-year benefit** (MEYG, fNfn)
 - Statistically significant improvement: We reran the N experiments with 20 repeats. We added the resulting confidence intervals in the updated version of Table 1. These results strongly support our claim about the importance of multi-year training.
 - Necessary for crop rotation: The crop rotation experiments would be impossible with single-year or single-crop simulations.

**Comparison to other crop RL envs** (GiWq, zvAw, MEYG)
 - Cropgym [41] is based on the PCSE implementation of LINTUL3. Therefore, it implements a simpler CGM than Cycles that does not model N dynamics and is single-year. PCSE allows for different crops but we could not find how to do so in Cropgym in its documentation.
 - DSSATgym [24] builds on CERES-Maize, which is multi-year. However, we could not find how to run DSSATgym for multiple-year experiments in the documentation. DSSATgym only supports maize and extending it to other crops is very challenging, according to the documentation.

@GiWq: therefore comparing to [24, 41] is not possible for experiments that span multiple years. Single-year experiments can be run but a comparison is challenging due to different observation spaces. The choice of environment for single-year studies depends on the underlying CGM and the research question (see Why Cycles).
May we ask for further details on the comparison the reviewer has in mind?

**Realistic scenarios** (MEYG, rNfn, GiWq)
We agree that farmers need to consider more factors than those presented here, e.g., the fixed costs associated with fertilization. Cyclesgym can easily implement these scenarios (see the “build_custom_environment” notebook). However, it is hard to quantify such costs as they depend on many factors. In Sec 6.1.1 of our revised version, we include new experiments investigating the performance of the RL agents as a function of such costs. These experiments show realistic fertilization patterns and still outperform the baselines.

We plan to perform further experiments with crop rotations that include winter cover crops. To engage early in a productive conversation, we have replied before completing these experiments. We will submit a new revised version when they are ready.

Finally, we highlight that Cyclesgym’s value resides in the pre-implemented scenarios and even more in its ease of expansion to create arbitrarily complex ones.

**Why Cycles?** (MEYG, GiWq, zvAw)
Different CGMs have different weaknesses and strengths (for a summary, see Di Paola, et al. “An overview of available crop growth and yield models for studies and assessments in agriculture”, 2015). Therefore, which one to use depends on the research question under investigation and there is no single best one.

Cycles’ advantages over other CGMs:
 - It is adept at modelling complex agricultural systems and the resulting interactions including crop rotations (Pravia et al., “Soil carbon saturation, productivity, and carbon and nitrogen cycling in crop-pasture rotations”, 2019) and polycultures (Burton et al.,  2021; dissertation), which remains difficult in other CGMs.
 - It simulates soil carbon-nitrogen dynamics accurately (Pravia et al., 2019), which makes it suitable for our research.

**Documentation** (GiWq, rNfn)
The documentation can indeed be improved. To start a productive conversation, we have replied before all the improvements are completed. To date, we added multiple markdown files to explain the most important aspects of cyclesgym. Moreover, we added more notebook examples for basic use cases. In the rest of the response period, we will add docstrings and expand the example section.

**Support for other CGMs**(GiWq, zvAw)
CGMs do not usually allow for interactive simulations where management decisions can be made during the simulated period based on observations. Therefore, they typically require dedicated wrappers to be used in an RL loop. Since different CGMs have different implementations, each one needs its own wrapper, which requires significant engineering effort to develop. Interoperability of CGMs is a large field of research (see Rosenzweig, et al. "The agricultural model intercomparison and improvement project (AgMIP): Protocols and pilot studies.",2013) that goes beyond the scope of this work.

**Code accessibility** (GiWq, yZVT, zvAw)
The email verification step was just for the review process. We always meant to make the code openly accessible upon publication. Due to the difficulties encountered by the reviewers, we opened the repository for the duration of the discussion period.

---

### Author Response · Authors · 2022-08-25
**Discussion on the revised version**

Dear reviewers, we would like to thank you again for your helpful feedback.

If you feel that our revised version addressed your concerns, please consider updating your score accordingly. Otherwise, we would be interested in hearing your opinion on the updated version of the paper to address new or old issues by the end of the rebuttal period.


@GiWq: We are currently running additional crop rotation experiments that include cover crops. Preliminary results are qualitatively similar to those included in the paper. We will add them as soon as they are ready.

---

### Meta-Review · Area_Chair_TcSE · 2022-09-06

**Recommendation:** Accept
**Confidence:** 4

**Metareview:**

This proposal introduces CylesGym, the first Reinforcement Learning (RL) benchmark targeted at long horizon decision making in agriculture. Crucially, while prior work addresses single-year decision making, CylesGym captures the long term effects that one year's crop has on future generations.

The benchmark is clearly highly relevant and opens up a new frontier for RL researchers, making it a valuable contribution to the field. Furthermore, the benchmark does a good job of highlighting interesting opportunities for RL method development, such as costly information gathering, and evaluating current algorithms compared to baselines.

There was an active discussion between the reviewers and authors of the benchmark which resolved the majority of issues raised in the initial reviews. As a result there is broad support across the reviewers for the paper. A lingering concern is the sim-to-real gap which is mentioned at a number of places in the paper but could be emphasised more.

Lastly, the paper is well written and the evaluation sound. I believe this benchmark will be welcome by the community but I recommend that the authors address the concerns regarding the writing raised by  Reviewer MEYG, in particular regarding the utility for the agriculture community. In general I do not believe that issues which can be addressed in writing should be a reason to reject but those concerns should be addressed for the final version.

---

### Decision · Program_Chairs · 2022-09-16

Accept